# Effects of Diazepam on Low-Frequency and High-Frequency Electrocortical γ-Power Mediated by α1- and α2-GABA_A_ Receptors

**DOI:** 10.3390/ijms20143486

**Published:** 2019-07-16

**Authors:** Julian I. Hofmann, Cornelius Schwarz, Uwe Rudolph, Bernd Antkowiak

**Affiliations:** 1Werner Reichardt Center for Integrative Neuroscience, Eberhard-Karls-University Tübingen, 72076 Tübingen, Germany; 2Department of Comparative Biosciences, College of Veterinary Medicine, University of Illinois at Urbana-Champain, Urbana, IL 61802-6178 USA; 3Department of Anesthesiology and Intensive Care, Experimental Anesthesiology Section, Eberhard-Karls-University Tübingen, 72072 Tübingen, Germany

**Keywords:** aminobutyric acid(A) receptors, GABA(A)-receptor subtypes, benzodiazepine, hypnotic, electroencephalogram, gamma-power

## Abstract

Patterns of spontaneous electric activity in the cerebral cortex change upon administration of benzodiazepines. Here we are testing the hypothesis that the prototypical benzodiazepine, diazepam, affects spectral power density in the low (20–50 Hz) and high (50–90 Hz) γ-band by targeting GABA_A_ receptors harboring α_1_- and α_2_-subunits. Local field potentials (LFPs) and action potentials were recorded in the barrel cortex of wild type mice and two mutant strains in which the drug exclusively acted via GABA_A_ receptors containing either α_1_- (DZα_1_-mice) or α_2_-subunits (DZα_2_-mice). In wild type mice, diazepam enhanced low γ-power. This effect was also evident in DZα_2_-mice, while diazepam decreased low γ-power in DZα_1_-mice. Diazepam increased correlated local LFP-activity in wild type animals and DZα_2_- but not in DZα_1_-mice. In all genotypes, spectral power density in the high γ-range and multi-unit action potential activity declined upon diazepam administration. We conclude that diazepam modifies low γ-power in opposing ways via α1- and α2-GABA_A_ receptors. The drug’s boosting effect involves α2-receptors and an increase in local intra-cortical synchrony. Furthermore, it is important to make a distinction between high- and low γ-power when evaluating the effects of drugs that target GABA_A_ receptors.

## 1. Introduction

Electroencephalographic activity in the high-frequency range (>20 Hz) is prominent during sensory processing and cognition. It is a signature of conscious cognitive functions and subject to modulation by clinically used benzodiazepines, which act by positive allosteric modulation of GABA_A_ receptors. Diazepam is a prototypical benzodiazepine that targets a family of GABA_A_ receptor subtypes containing the α1-, α2-, α3- or α5-subunit [1]. These subunits display different expression patterns in the central nervous system [2] and serve different physiological functions [3]. However, their individual contribution to the overall effects of benzodiazepines on high-frequency brain activity is unknown. To uncover actions of diazepam produced by only a single subtype of the GABA_A_ receptor, we used mice carrying knockin mutations in 3 out of 4 benzodiazepine-sensitive subunits (Figure 1). Mutations of a histidine residue into an arginine in the α-subunits dramatically reduced diazepam binding without attenuating receptor activation by the natural agonist γ-aminobutyric acid [4]. The effects of diazepam were investigated in two mutant mouse strains in which the drug exclusively acted via GABA_A_ receptors containing either α1- (DZα1-mice) or α2-subunits (DZα2-mice) [5,6,7]. In DZα1-mice, point mutations in the α2-(H101R), α3-(H126R) and α5-(H105R) subunits rendered the respective GABA_A_ receptor subtypes resistant to diazepam. In DZα2-mice, the corresponding mutations were located in the α1-(H101R), α3-(H126R) and α5-(H105R) subunits. It is important to note that these point mutations do not compromise the physiological functions of GABA_A_ receptors. Thus, a difference between wild type, DZα1- and DZα2-mice only develops in the presence of benzodiazepines. Receptors containing α1- and α2-subunits define a major and a minor subpopulation, as they are abundant in 60% and 20% of brain GABA_A_ receptors, respectively [1]. In the present work, we recorded local field potentials and actions potentials in the mouse barrel cortex. We observed that diazepam reduced spontaneous action potential firing of cortical neurons via both, α1- and α2-GABA_A_ receptors. However, enhancement of spectral power in the low γ-range and an increase in intra-cortical synchrony were observed in wild type and DZα2- but not in DZα1-mice. Our findings suggest that diazepam is boosting low γ-electroencephalographic activity and intra-cortical synchronicity predominantly via α2-GABA_A_ receptors.

## 2. Results

### 2.1. Diazepam’s Effects on Low- and High γ-Power

Classical benzodiazepines alter electrocortical activity in the high frequency range, but it is largely unknown which subtypes of the GABA_A_ receptor are involved. In order to elucidate the roles of α1- and α2-GABA_A_ receptors, we monitored spontaneous local field potentials in the barrel cortex of wild type, DZα1- and DZα2-mice. The position of recording electrodes is indicated in Figure 2A. In Figure 2B representative LFP traces are displayed. The signals shown in the figure were derived from DZα1- and DZα2-, and wild type-mice under drug-free conditions (black lines) and after diazepam administration (10 mg/kg body weight, intraperitoneal injection, blue lines). Periods in which the mouse moved were eliminated from the data.

The upper panels in Figure 3A–C show power spectral densities of LFP recordings at frequencies between 20 and 90 Hz in the three genotypes. All spectra are monotonically decaying with higher frequency, but the ones calculated from the diazepam condition deviate from their respective drug-free controls. Only in DZα1-mice (Figure 3A) did diazepam have a depressing effect across the entire frequency range. In DZα2-mice (Figure 3B) the drug boosted low-γ power (<50 Hz), but decreased high γ-power (>50 Hz). In wild types (Figure 3C), diazepam-induced power enhancement was apparent only at frequencies <30 Hz. The difference in crossover points (between the drug-free and diazepam condition) in DZα2- (at ~50 Hz) and wildtype-mice (at ~30 Hz) is explained by diazepam’s simultaneous action at α1- and α2-GABA_A_ receptors in wild types. Thus, α1-receptor-mediated inhibition of low γ-power is overcompensated by α2-receptor-mediated power enhancement at frequencies below 30 Hz.

Since data were not normally distributed, we chose a non-parametric approach to perform statistical analysis (Figure 3A–C, lower panels). The area under the receiver-operated curve (AUC) is a measure of the difference between two populations. It was applied to the power spectra readings in each frequency bin (resolution 1 Hz), obtained under diazepam versus the drug-free condition. An AUC value of 0.5 indicates no effect (complete overlap of LFP recordings in the absence and presence of diazepam) while an AUC value of 0 or 1 indicates that the two distributions are completely disjunct, with diazepam giving rise to the higher power if the AUC value is 1, and to the lower power if the AUC value is 0. The lower three graphs in Figure 3 contain AUCs from each animal (thin black lines) surrounded by the [0.025, 0.975] prediction intervals (or 95% confidence intervals, gray bands), as well as the grand average across animals (thick black lines). Power distributions are, at an α-error level of *p* < 0.05, different between the diazepam and drug-free condition if the gray bands do not include the thick horizontal black line at AUC = 0.5. When applying this approach, we found that in all tested animals diazepam significantly reduced LFP power at frequencies in the high γ-band between 50 and 90 Hz (AUC < 0.5). However, diazepam’s actions in the low γ-band (between 20 and 50 Hz) were different. In DZα2-mice and wildtypes, the drug enhanced low γ-power, whereas the inverse effect was evident in DZα1-mice.

When comparing drug-free recordings with those carried out after injecting the carrier Lipofundin, AUC prediction intervals included the value of 0.5 across the entire frequency range, indicating that the carrier and the injection procedure did not have a significant effect (data not shown). In summary, the results presented in Figure 3 support the view that diazepam increases low γ-power via positive modulation of α2-containing GABA_A_-receptors, while positive modulation of α1-containing GABA_A_-receptors reduces low γ-power. Unlike in the triple knockin mice, diazepam potentiates α1- and α2-receptors in wild types at the same time. In these animals the α2-mediated effect dominates at frequencies below 30 Hz. Effects of diazepam in the high γ-band were different. In contrast to low γ-power, diazepam depressed in all genotypes high γ-power above 50 Hz.

### 2.2. Diazepam Curtails Action Potential Firing in All Genotypes

Next, we investigated potential cellular mechanisms causing the changes in γ-power produced by diazepam. Specifically, we raised the question of whether the drug’s action on low and high γ-power involves changes in the discharge rate of cortical neurons or, alternatively, changes in correlated network activity. To answer this question, we first quantified the frequency of spontaneous action potential firing in the presence and absence of the drug. Multi-unit spike rates varied strongly between recording channels and drug conditions (drug-free: 1–110 spikes/s; diazepam: 0.1–65 spikes/s). We related the firing rates monitored under drug-free conditions (before diazepam administration and after recovery from the treatment) to those observed in the presence of the drug. Each point displayed in Figure 4 represents recordings from a single electrode. In all genotypes, the vast majority of circles fall into the grey shaded area below the diagonal and marked by the dotted line, indicating lower discharge rates in the presence of diazepam. Thus, diazepam attenuated action potential firing in all genotypes.

### 2.3. Diazepam Increases Correlated Firing in the Low γ-Range via α2- but Not α1-GABA_A_ Receptors

Diazepam-induced decline of discharge rates, which was evident in all genotypes, does not provide an explanation for the observation that the drug enhanced low γ-power in DZα2- and wild type, but not in DZα1-mice. Therefore, we tested the hypothesis that only in the former two genotypes the drug increased correlated neuronal activity in the low γ-range. LFP synchrony was quantified by calculating the element-wise Pearson correlation coefficient between pairs of LFP recordings obtained from the four recording electrodes in each animal. To this end, a frequency window (width 20 Hz) was moved in steps of 10 Hz across the frequency range 20 to 90 Hz (the windows thus covered frequencies from 10 to 100 Hz). The pair of LFPs was band pass filtered at edge frequencies determined by each frequency window, and for each of these frequency bands the synchrony was assessed. Figure 5 shows the effect of diazepam as the ratio of average correlations under diazepam and drug-free ($q={\bar{r}}_{diazepam}/{\bar{r}}_{drugfree}$). Wild type- and DZα2-animals stand out with correlation ratios ranging from 0.7 to 1.6 while correlations obtained in DZα_1_-mutants were comparably low. Correlations of LFPs at frequencies <50 Hz in WT and DZα_2_-animals were enhanced under diazepam (mean ± s.d.: wild types, *q* = 1.09 ± 0.13; DZα1-mice, *q* = 1.03 ± 0.05; DZα_2_-mice, *q* = 1.18 ± 0.13; *t*-test for independent samples: wild types vs. DZα_1_-mice and DZα_2_-mice vs. DZα_1_-mice both *p* << 0.01) whereas correlations at higher frequencies were largely unaffected. These results support the view that α_2_-GABA_A_ receptor-mediated enhancement of low γ-power is due to an increase in local cortical synchrony.

## 3. Discussion

### 3.1. Modulation of Low γ-Power via α2-GABA_A_ Receptors

Increased spectral power density in the β- and low γ-frequency band is a well-known component of the EEG spectral signature produced by benzodiazepines [13,14]. We found that modulation of α2-GABA_A_ receptors is sufficient for increasing low γ-power. Furthermore, diazepam-induced augmentation of low γ-power in DZα2-mice was associated with enhanced correlated firing of cortical neurons but not with increased discharge rates. These findings suggest that benzodiazepines are boosting low γ-band power predominantly via the enhancing of synchronized firing of cortical neurons, but not by increasing discharge rates. The conclusion that benzodiazepine-induced enhancement of low γ-power involves α2-GABA_A_ receptors is in good agreement with the study of Christian and coworkers [14]. These researchers reported that chemically distinct compounds acting as positive allosteric modulators at α2/3-, but not α1-GABA_A_ receptors, caused robust enhancement in spectral power in the β- and low γ-band (12–50 Hz) when administered at non-sedating doses. Another study in humans supports the association between α2-subunits and β-power increments: Polymorphisms in GABRA2, the gene coding for the α2-subunit, have been shown to be tightly connected, with increased spectral power in the β-frequency band in families with multiple alcoholics [15].

### 3.2. Diazepam Curtails High γ-Power and Action Potential Firing in All Genotypes

Unlike the genotype-specific actions of diazepam observed in the low γ-frequency band, spectral power in the high γ-frequency range was decreased by the drug in Dzα1-, Dzα2-, as well as in wild type mice. Furthermore, diazepam reduced spontaneous multi-unit action potential firing in all genotypes. The similar changes in multi-unit firing and high γ-power suggest a close relationship between these signals. Moreover, depression of high γ-power and/or multi-unit firing seems to be a common feature of sedative drugs, largely independent of their molecular mechanism of action. In rats, high γ-power was attenuated by the GABAergic anaesthetic propofol [16] and by ketamine [17], which is thought to act predominantly via NMDA-receptors and HCN1-channels [18]. Likewise, volatile anesthetics, displaying a spectrum of molecular actions that considerably deviates from those of propofol and ketamine, also decreased high γ-power [19] and multi-unit discharge rates [20] in rats. Thus, the changes in high γ-power and multi-unit firing rates, which seem to be produced by almost all sedative drugs investigated so far, cannot be used as a biomarker for a specific molecular drug target.

### 3.3. Related Studies on Single Knockin Mice

Irene Tobler’s group investigated diazepam’s effects on the EEG in knockin mice, in which only a single subtype of benzodiazepine-sensitive GABA_A_ receptors was rendered insensitive to the drug. These authors evaluated changes of EEG spectra within a frequency range of 1–25 Hz. In wild types, diapezam increased spectral power density in the β-band (12–25 Hz). However, in α2(H101R)-mutants (according to the notation used in the present study these are Dzα 1+3+5-mice since diazepam acts via α1-, α3- and α5-GABA_A_ receptors) diazepam failed to increase β-power, suggesting that this effect is mediated by α2-GABA_A_ receptors [21]. Diazepam-induced changes in the β-frequency band observed in the present study (data not shown) were fully consistent with these findings. These observations, together with the results displayed in Figure 3, indicate that diazepam, via targeting α2-GABA_A_ receptors, boosts spectral power in the β- and, in addition, in the low γ-band. Remarkably, diazepam-induced enhancement of β-power in α1(H101R)-mice (our notation: Dzα 2+3+5-mice) was significantly stronger than what was observed in wild types [13], leading to the conclusion that spectral power density in the β-band of the EEG is reduced by benzodiazepines via α1-GABA_A_ receptors. Thus, β-power is modified by α1- and α2-GABA_A_ receptors in opposing ways. Again, these findings compare well to what we observed in Dzα1- and Dzα2-mice in the low γ-band. It is worth mentioning that Kopp and coworkers did not observe any differences in the effects of diazepam in wild type and α3(H126R) knockin mice (our notation: Dzα 1+2+5-mice) within a frequency range of 1–25 Hz, indicating that α3-GABA_A_ receptors do not provide a major contribution to the characteristic EEG fingerprint of benzodiazepines [22]. However, diazepam-induced modulation of cortical activity can be behavior-specific [23,24], raising the possibility that α3-dependent effects are apparent only during behavioral states not investigated so far. Besides α1-, α2-, and α3-subunit containing GABA_A_ receptors, diazepam also binds to α5-receptors. Selective pharmacological modulation of α5-receptors significantly altered spontaneous action potential firing of cortical neurons in vitro [25]. Furthermore, a recent study evaluated changes in EEG activity in vivo caused by a drug acting as a negative allosteric modulator on α5-receptors [26]. This agent increased γ-power (30–80 Hz) but did not change power in the δ-, θ-, α-, and β-band. Figure 6 summarizes the conclusions based on studies on single and triple knockin mutant mice.

### 3.4. Diazepam-Induced Changes in Motor Activity

Diazepam’s opposed actions on β- and low γ-EEG power involving α1- and α2-receptors (Figure 6) suggest that these receptor subtypes also may affect behavior in opposing ways. The study of Ralvenius and coworkers showed that in Dzα1-mice, diazepam significantly decreased locomotor activity [6]. Contrastingly, the drug strongly enhanced locomotor activity in Dzα2-mice. Furthermore, the authors assessed diazepam-induced changes in locomotor activity in quadruple point-mutated mice. They found significant and dose-dependent impairment of locomotor activity, muscle strength, and motor coordination, which were absent in vehicle-treated animals. These behavioral actions of diazepam remaining in the quadruple-point-mutated mice may involve a low-affinity benzodiazepine-binding site [27,28]. There is evidence in the literature that benzodiazepines, at clinically relevant concentrations, can also enhance neurosteroidogenesis and GABAergic inhibition via binding to the translocator protein (18 kDa) [29]. However, promoting the synthesis of endogenous neurosteroids has been linked to anxiolysis with no effect on motor activity in rodents [30]. Interestingly, the volatile anesthetics halothane and isoflurane, and the intravenous anesthetic propofol, all of which enhance GABA_A_ receptor-mediated synaptic inhibition when administered at sedative concentrations [20,31], also increase locomotor activity [32]. In the latter study, inactivation of the medial septum by local injection of muscimol abolished behavioral excitation, suggesting that this effect is mediated at least in part by the septohippocampal system.

### 3.5. Limitations of the Present Study

The present study utilized global knockin mice, which is the best available animal model for elucidating functional differences between subpopulations of GABA_A_ receptors in vivo. However, our novel finding that α1- and α2-subunits are shaping low γ-power in different ways is raising additional questions concerning the cellular and network mechanisms that are involved, as the location on specific neuronal populations and circuits is critical for the effects mediated by GABA_A_ receptor subtypes, which has been referred to as “circuit pharmacology” [33]. Our experiments monitored cortical activity and intra-cortical actions of diazepam. In cortical networks, α1- and α2-subunits are abundant and therefore these receptor subtypes likely contribute to the drug’s effects reported here. However, as knockin mutations were global and α1- and α2- subunits are also present in the septohippocampal system, arousal nuclei, sleep-promoting pathways, and the amygdala, sub-cortical actions of diazepam are also likely to come into play [3]. Conditional knockin animals may provide a tool for addressing this issue in the future, as may new techniques that allow cell type-specific drug action such as drugs acutely restricted by tethering [34].

## 4. Materials and Methods

### 4.1. Animals

High affinity binding of benzodiazepines to GABA_A_ receptors requires the presence of an α1-, α2-, α3-, or α5-subunit [1]. The mouse lines used here had homozygous knockin mutations in three out of the four mentioned subunits. Our notation is DZαX-mice, indicating animals in which diazepam exclusively acts via subunit X-containing GABA_A_ receptors. DZα2-mice were crossbred from animals carrying a mutation of either α1-H101R- [10], α3-H126R- [11], or α5-H105R- [12] GABA_A_-receptor subunits. The histidine to arginine point mutations render the respective subunits insensitive to modulation by diazepam. The second mutant, DZα1-mice, contained mutations in the α2-H101R- [11], α3- H126R-, and α5- H105R- GABA_A_-receptor subunits. The Dzα1-mice have previously been referred to as “HRRR” and the Dzα2-mice as “RHRR” [6]. Wild type mice were, as the two triple mutants, on the 129X1/SvJ background.

Eleven male mice aged 15–35 weeks and weighing 28 to 35 g were used in this study (wild type: *n* = 4; DZα2-mice: *n* = 4; DZα1-mice: *n* = 3). The animals were maintained on a regular 12-hour light/12-hour dark cycle. Food and water were provided ad libitum. All experimental and surgical procedures were approved by the local German authorities (Regierungspräsidium Tübingen, code N8/09, date of approval: 30 November 2009).

### 4.2. Implantation of Recording Electrodes

Electrode implantation was done as described previously [35,36]. Briefly, anesthesia was induced with 100 mg/kg ketamine (WDT, Garbsen, Germany) and 15 mg/kg xylazine (Rompun, Bayer AG, Leverkusen, Germany) and maintained with isoflurane (cp-pharma, Burgdorf, Germany) inhalational anesthesia at levels from 1–1.5%, abolishing pain reflexes. Animals were placed in a stereotaxic apparatus, the skull prepared and a burr hole drilled. Without removing the dura, a 2-by-2 array of platinum-tungsten micro-electrodes (interelectrode spacing: 300 µm; impedance <1 MΩ) was implanted in barrel cortex at a depth of 500 µm (P 1.8 and L 3.5 with reference to bregma). A silverball electrode placed on the surface of the cerebellar hemispheres was used as reference, while another one was used as signal ground. The head cap was built up with dental cement (Tetric EvoFlow, Ivoclar Vivadent AG, Schaan, Liechtenstein) and held a screw for head fixation (thread diameter: 3 mm). Careful hemostasis, disinfection, and wound suture completed the surgery. Postoperative care included warmth, antibiotic (7 days; Baytril/enrofloxacin, Bayer AG, Leverkusen, Germany), and pain treatment (2 to 3 days; Carprofen, Pfizer Animal Health, New York, NY, USA).

### 4.3. Drug Administration and Electrophysiological Measurements

Habituation of the mice to head fixation was done exactly as described previously for rats [36]. All recording sessions were performed in the morning between 9 am and 12 noon. Each animal was tested under three conditions (sequence random for each animal): One session with injection of diazepam (called “diazepam” condition), another with the carrier Lipofundin (called “carrier” condition) and 3 to 5 sessions without a drug (called “drug-free” condition). Diazepam sessions were followed by a pause of at least one week to assure the drug’s complete metabolization. Before the recording session, a single injection of diazepam (10 mg/kg body weight; Sigma-Aldrich, St Louis, MO, USA), or the carrier lipofundin (same volume as diazepam; Lipofundin® MCT 20%, B. Braun Melsungen AG, Germany) was administered (i.p.). In the drug-free sessions no injection was performed. For electrophysiological monitoring the mice were then head fixed in a restrainer box. The electrophysiology session started exactly 10 min after the injection and lasted between 12 to 35 minutes. Electrophysiological signals were recorded, band-pass filtered between 1 and 5000 Hz, amplified (gain: 2000) and digitized (16-bit depth) (USB-ME128-PGA-System; MultiChannelSystems, Reutlingen, Germany). Movement of the animal was recorded using a touch sensitive piezo element, placed under the forepaws.

### 4.4. Data Structure

The recorded voltage traces were subdivided into intervals of 1.024 s duration (comprising 2048 data points at a sampling frequency of 2 kHz). Intervals in which the mouse moved were eliminated from the data by the experimenter examining the signal of a piezo element. The final data set included 201–585 movement free trials under diazepam condition, 89–233 under carrier condition, and 217–537 under drug-free condition, per animal (see representative examples in Figure 2). The data analysis was realized using custom-written software in Matlab (version 2010a, The MathWorks™, Natick, MA, USA). LFP data were obtained by low-pass filtering the data at an edge frequency of 200 Hz (7th order butterworth filter) followed by downsampling to 2 kHz. Multi-unit spikes were extracted from band-pass filtered data (300–5000 Hz) using a window discriminator. All spike waveforms were biphasic (negative peak first) as typically encountered with extracellular spikes sampled in the neocortex.

#### LFP Analysis

The power of the LFP was calculated using a discrete Fourier transformation on each 1.024 s interval. The power spectrum was filtered with a moving average using a window of 5 Hz moved across the spectra in steps of 1 Hz. The upper panels in Figure 3 show data from one channel per animal of each group (interval-averaged samples).

To compare the power at each frequency bin of 1 Hz, we calculated the effect size of the power distributions obtained for diazepam, and carrier sessions using the receiver operating characteristics (ROC) as described by Hentschke and Stüttgen [37]. The area under the ROC curve is interpreted as the probability with which an ideal observer correctly classifies an interval of LFP as originating from one or the other measured distribution averaged over all possible classification criteria. An AUC value of 0.5 indicates no effect, while an AUC of 0 and 1 indicate complete separation of the distributions (perfect classification possible). For statistical testing of the AUC values, 2000 bootstrap estimates were calculated to achieve a 95% prediction interval (gray bands in Figure 3, lower panels). 

For synchrony measures of LFP signals in different channels, the channels were first band-pass filtered by a 7th order butterworth filter (pass band 10 to 100 Hz). Pearson correlation coefficient r calculated from each pair of synchronous LFP recordings was employed as a measure of synchrony, i.e., only synchrony without delays was considered. The overall effect of diazepam on synchrony reported in Figure 5 was quantified as the ratio of the average coefficient $\bar{r}$ obtained from pairs of channels in all recording of diazepam and drug-free conditions (${\bar{r}}_{diazepam}/{\bar{r}}_{drugfree}$).

## 5. Conclusions

Diazepam alters spectral power in the low γ-frequency band via α1- and α2-GABA_A_ receptors in opposing ways. Positive modulation of α1-receptors, which define the major sub-population of GABA_A_ receptors in the cerebral cortex, decreases low γ-power. However, in wildtype mice, this effect is overwritten by diazepam’s enhancing action, involving α2-GABA_A_ receptors. The increase in low γ-power produced by diazepam may reflect increased intra-cortical synchrony. In the high γ-frequency band, diazepam reduced spectral power via both α1- and α2-GABA_A_ receptors. 

## Figures and Tables

**Figure 1 ijms-20-03486-f001:**
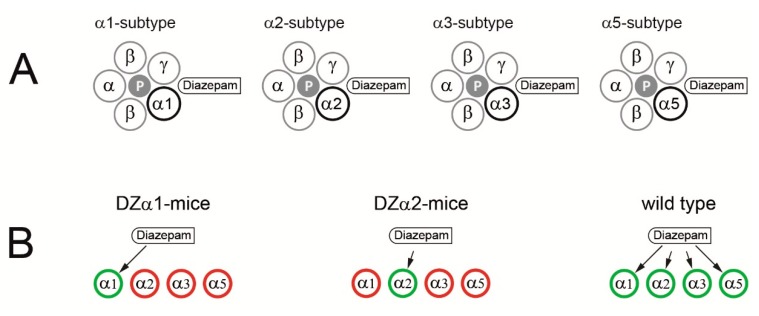
(**A**) GABA_A_ receptors assemble from five protein subunits. Most receptors in the brain are composed of two α-, two β- and a γ-subunit arranged γβαβα counterclockwise around a central pore (P) as viewed from the extracellular side of the neuron [8]. Benzodiazepines bind to an extracellular intersubunit site formed by an α1-, α2-, α3- or an α5-subunit and a γ-subunit [9]. Hence, these α-subunits define four distinct benzodiazepine-sensitive receptor subtypes. (**B**) Point mutations in the α-subunits render GABA_A_ receptors insensitive to diazepam [10,11,12]. In DZα1-mice, diazepam exclusively acts via α1-containing GABA_A_ receptors (green circles), because α2-, α3- and α5- subunits carry this type of knockin mutation (red circles). In DZα2-mice, diazepam is only active at α2-GABA_A_ receptors. In wildtype animals, multiple subtypes of the GABA_A_ receptor contribute to the effects produced by diazepam.

**Figure 2 ijms-20-03486-f002:**
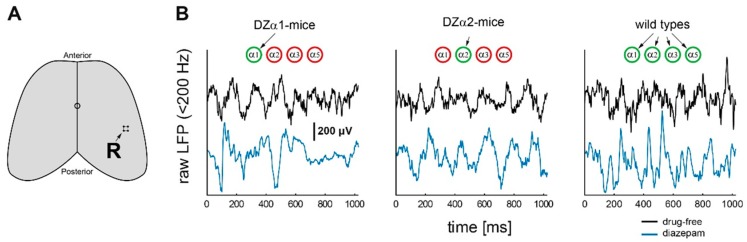
(**A**) Time series of electrocortical activity were recorded with an array of 4 electrodes which was chronically implanted in the cortex at a depth of 500 μm. R: The site of recording (bregma P 1.8 and L 3.5). (**B**) Representative local field potential signals under drug-free conditions (black traces) and after intraperitoneal administration of diazepam at a concentration of 10 mg/kg body weight (blue traces). Signals were band-pass filtered, using cutoff frequencies of 1 and 200 Hz, respectively.

**Figure 3 ijms-20-03486-f003:**
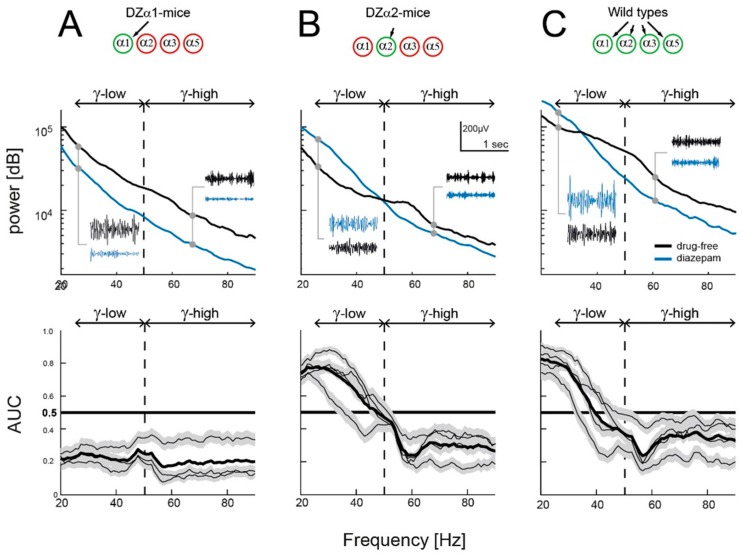
Effect of diazepam on γ-power in (**A**) DZα1-, (**B**) DZα2- and (**C**) wild type-mice. Upper panels: Trial-averaged power spectra calculated for a frequency range of 20 to 90 Hz. One representative animal for each group of genotypes is shown (number of trials for drug-free/diazepam: DZα1 = 282/314; DZα2 = 340/316; wildtype = 294/437). Small insets: raw local field potentials (LFPs) were filtered to pass frequencies between 20–50 Hz (low γ-range) and 50–90 Hz (high γ-range), respectively. Lower panels: Effects of diazepam on LFP power as the area under the receiver operating characteristics curve (AUC) for each mouse (thin black lines; gray field: [0.025 0.975] bootstrapped prediction interval) and averaged across mice for all genotypes (thick lines). An AUC of 0.5 indicates the absence of an effect (horizontal black line).

**Figure 4 ijms-20-03486-f004:**
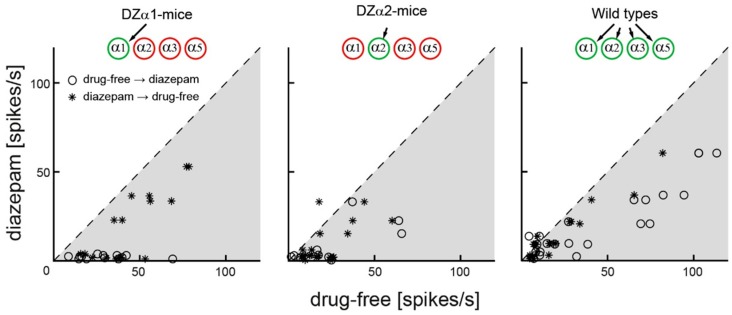
Spikes per second under diazepam (ordinate) plotted versus spikes under drug-free conditions (abscissa) for all three genotypes. The broken lines represents identity of discharge rates and the grey-shaded area a decrease in action potential activity under diazepam. Diazepam attenuated spontaneous action potential firing in all genotypes. This effect is independent of whether the drug-free session was recorded before (circles) or after the diazepam session (asterisks).

**Figure 5 ijms-20-03486-f005:**
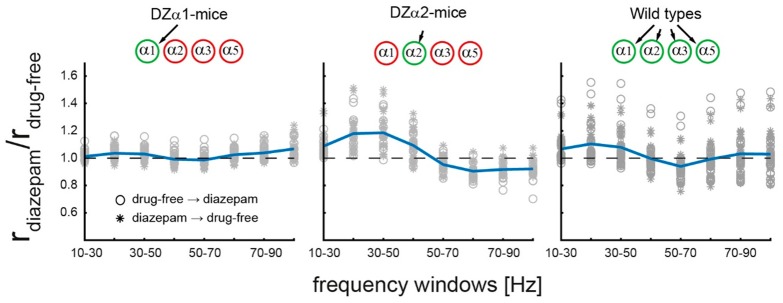
Correlation of LFP signals recorded on all electrodes of the 2 × 2 array. The relative correlation is plotted for different frequency bands, using windows of 20 Hz in size. LFP signals were band-pass filtered. Cutoff frequencies were 10 and 100 Hz, respectively. Blue lines: Mean ratio across sessions and animals in one genotype. Symbols denote the sequence of the drug-free and diazepam session (Open circles: drug-free session first, Asteriks: Diazepam session first). In DZα2-animals, enhancement of correlation in the low γ-frequency range is most pronounced, but absent in DZα1-mice.

**Figure 6 ijms-20-03486-f006:**
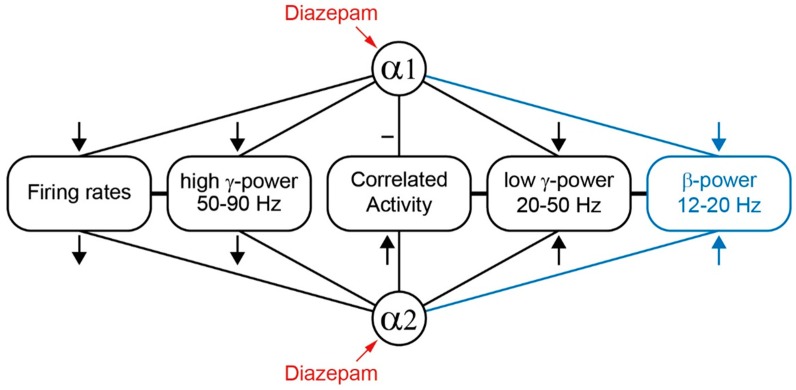
Effects of diazepam on electrocortical activity mediated by α1- and α2-GABA_A_ receptors. Diazepam’s actions on β-power (blue) were evaluated in studies on single knockin mice [13,21]. From these and our study it is concluded that positive modulation of α1-GABA_A_ receptors reduces β-, low γ-, high γ-power and spontaneous action potential firing of cortical neurons as indicated by downwardly directed arrows, leaving intracortical correlated activity unchanged. Positive modulation of α2-GABA_A_ receptors reduces firing rates and high-γ power, supporting the idea of a close relation between these signals, as indicated by the horizontal lines between boxes. Furthermore, modulation of α2-GABA_A_ receptors increases β-, low γ-power and correlated activity, supporting the hypothesis that these effects are caused by increased intracortical synchrony (indicated by horizontal lines between boxes). Remarkably, β- and low γ-power are modified via α1- and α2-GABA_A_ receptors in opposing ways.

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
