# Peer review of "Effects of Diazepam on Low-Frequency and High-Frequency Electrocortical γ-Power Mediated by α1- and α2-GABAA Receptors"

_ijms, 2019, doi:10.3390/ijms20143486_

Round 1
Reviewer 1 Report
This is a clearly written and well illustrated paper that reports on the effects of targeted knockouts of different GABAA receptor subtypes on the effects of benezodiazepines. Interesting findings are reported for the differential involvement of these receptor subtypes in benezodiazepine effects on EEG and neuronal discharge. These results further establish the selectivity of GABAergic agents and point the way to further characterization of these important inhibitory receptors. I have only a couple of minor issues:
“largely independent on their molecular mechanism” should be: largely independent of their molecular mechanism
“It is worth to mention that Kopp” should be: It is worth mentioning that Kopp
“GABAA receptors in vivo” should be: GABAA receptors in vivo.
“Our experiments monitor cortical activity” should be: Our experiments monitored cortical activity
“receptor subtypes possibly contribute to the” should be: receptor subtypes likely contribute to the
Reviewer 2 Report
In this manuscript, the authors reveal the action of diazepam produced by a single subtype of the GABAA receptor using mice carrying knock-in mutation especially in a1-DZa1 & a2- DZa2 mice. This is based on the idea that these points mutation do not compromise the physiological function of GABAA receptors but difference develops only in the presence of benzodiazepines. They use diazepam to test this idea in mutant mice and observed that diazepam reduced spontaneous action potential firing of cortical neurons via both, a1- and a2-GABAA receptors but enhancement of spectral power in the low g-range and intra-cortical synchrony were observed in wild type and DZa2 but not in DZa1-mice. Understanding the role of the subpopulation of GABAA receptor on circuit pharmacology is important and this data adds to significant literature on the subject. I would recommend this research article for publication.
